# Effects of *Moringa oleifera* Seed Oil on Cultured Human Sebocytes In Vitro and Comparison with Other Oil Types

**DOI:** 10.3390/ijms241210332

**Published:** 2023-06-19

**Authors:** Christos C. Zouboulis, Amir M. Hossini, Xiaoxiao Hou, Chaoxuan Wang, Karsten H. Weylandt, Anne Pietzner

**Affiliations:** 1Departments of Dermatology, Venereology, Allergology and Immunology, Staedtisches Klinikum Dessau, Brandenburg Medical School Theodor Fontane and Faculty of Health Sciences Brandenburg, 06847 Dessau, Germany; 2Division of Medicine, Department of Gastroenterology, Metabolism and Oncology, University Hospital Ruppin-Brandenburg, Brandenburg Medical School and Faculty of Health Sciences Brandenburg, 16816 Neuruppin, Germany; 3Division of Psychosomatic Medicine, Medical Department, Campus Benjamin Franklin, Charité-Universitaetsmedizin Berlin, 12203 Berlin, Germany

**Keywords:** *Moringa oleifera*, Moringa seed oil, olive oil, sunflower oil, free fatty acids, linoleic acid, oleic acid, human sebaceous gland cell line SZ95

## Abstract

The seeds of *Moringa oleifera* (horseradish tree) contain about 40% of one of the most stable vegetable oils (Moringa seed oil). Therefore, the effects of Moringa seed oil on human SZ95 sebocytes were investigated and were compared with other vegetable oils. Immortalized human SZ95 sebocytes were treated with Moringa seed oil, olive oil, sunflower oil, linoleic acid and oleic acid. Lipid droplets were visualized by Nile Red fluorescence, cytokine secretion via cytokine antibody array, cell viability with calcein-AM fluorescence, cell proliferation by real-time cell analysis, and fatty acids were determined by gas chromatography. Statistical analysis was performed by the Wilcoxon matched-pairs signed-rank test, the Kruskal–Wallis test and Dunn’s multiple comparison test. The vegetable oils tested stimulated sebaceous lipogenesis in a concentration-dependent manner. The pattern of lipogenesis induced by Moringa seed oil and olive oil was comparable to lipogenesis stimulated by oleic acid with also similar fatty acid secretion and cell proliferation patterns. Sunflower oil induced the strongest lipogenesis among the tested oils and fatty acids. There were also differences in cytokine secretion, induced by treatment with different oils. Moringa seed oil and olive oil, but not sunflower oil, reduced the secretion of pro-inflammatory cytokines, in comparison to untreated cells, and exhibited a low n-6/n-3 index. The anti-inflammatory oleic acid detected in Moringa seed oil probably contributed to its low levels of pro-inflammatory cytokine secretion and induction of cell death. In conclusion, Moringa seed oil seems to concentrate several desired oil properties on sebocytes, such as high content level of the anti-inflammatory fatty acid oleic acid, induction of similar cell proliferation and lipogenesis patterns compared with oleic acid, lipogenesis with a low n-6/n-3 index and inhibition of secretion of pro-inflammatory cytokines. These properties characterize Moringa seed oil as an interesting nutrient and a promising ingredient in skin care products.

## 1. Introduction

*Moringa oleifera* (horseradish tree) belongs to the ben-nut-tree family (*Moringaceae*). Originally, the tree comes from the Himalayan region in northwest India, but it now grows worldwide in the tropics and subtropics—especially in countries of Africa, the Arabic peninsula, Southeast Asia, South America, and the Caribbean islands—and has been intensively cultivated for decades in southern India. An increase in cultivation can also be observed—albeit very slowly—in Africa. In Africa, the tree is preferably used as a natural remedy in a wide range of applications [1]. In poor regions of Africa, *Moringa oleifera* and *Moringa stenopetala* are cultivated as a primary food source, as the plants are suitable for combating malnutrition in these regions.

All plant parts of the horseradish tree are used in the local traditional medicine of India, Sri Lanka, Java and Africa. The juice is used to stabilize blood pressure. Leaves have an anti-inflammatory effect. The roots are used to cure rheumatic complaints. The alkaloids spirochine and moringine contained in the root have a bactericidal activity, which is why they are used as an antibiotic and in biological plant protection [2]. Currently, the anti-inflammatory and immunomodulatory properties of *Moringa oleifera* are under investigation [3]. 

As an inexpensive and simple method, water is purified using *Moringa oleifera* seeds in areas of extreme poverty with poor access to drinking water [4]. A barrel full of turbid river water can be clarified with 200–300 mg seed powder if it is stirred slowly and evenly for 15–20 min. In this process, the suspended matter and bacteria are flocculated by the seed powder and thus sink to the bottom. 

Moreover, the seeds contain about 40% of their weight as oil. Seed pressing produces one of the most stable vegetable oils (Moringa seed oil), which has a long shelf life and does not go rancid. The long shelf life of the oil is based on its higher stability against oxygen due to the low content of polyunsaturated fatty acids (PUFA), whose double bounds are very susceptible to oxidation [5]. Moringa seed oil is rated among the “high oleic oils” with a high monounsaturated fatty acids (MUFA)/PUFA ratio, comparable to olive oil; these are associated with a reduced risk of all-cause mortality, cardiovascular events and stroke [6]. In addition to the dominant fatty acids, such as oleic acid, palmitic acid or palmitoleic acid, it contains other bioactive substances, such as stigmasterol or tocopherols, which have antioxidant effects [7]. In various studies, Moringa seed oil showed bioactivity against Gram-positive (*Staphylococcus aureus* and *Bacillus subilis*) and Gram-negative bacteria (*Pseudomonas aeruginosa* and *Escherichia coli*), as well as antiplasmodial activity [6]. Some fatty acids and compounds contained in the oil may exert pro- and anti-inflammatory effects [8,9]; e.g., high concentrations of linoleic acid (C18:2n-6), an essential n-6-fatty acid, may protect against the development of inflammatory skin lesions [10], but it might also exhibit pro-inflammatory properties under different circumstances [11]. 

Lipid-producing cells (such as adipocytes and sebocytes) can integrate ingested free fatty acids into their own lipids. In the skin, this role is taken over by the sebaceous gland. The most obvious function of the sebaceous gland cells is the secretion of sebum [12]. Sebum contains a lipid mixture of triglycerides, squalene and wax esters, as well as cholesterol esters and possibly some free cholesterol, most of which are synthesized de novo by the sebaceous glands [13]. Human sebocytes were shown to produce the same amount of lipids in vitro after incubation with linoleic acid and palmitic acid (16:0), but their effects on sebocyte inflammatory signaling were strikingly different [14]. 

The immortalized human sebaceous gland cell line SZ95 model was developed to significantly reduce the number of animal experiments in the fields of developmental biology, lipogenesis and pharmaceutical and cosmetic compound testing [15].

In this work, we assessed the effects of Moringa seed oil on human SZ95 sebocytes and compared them with the effects of other vegetable oils, namely olive oil and sunflower oil, as well as with the pure fatty acids linoleic acid and oleic acid. Our aim was to assess any regulatory properties of Moringa seed oil on the physiology of human sebocytes in order to preclinically evaluate a possible application of this oil as an ingredient of skin care products and as a topical drug vehicle in addition to its use as a nutrient in Africa.

## 2. Results

### 2.1. Fatty Acid Content in the Vegetable Oils

The distribution of the main fatty acids in the vegetable oils studied is shown in Table 1.

### 2.2. Moringa Seed Oil Stimulates Sebaceous Lipogenesis

We studied the effect of Moringa seed oil, olive oil and sunflower oil but also of linoleic acid and oleic acid on inducible lipid production in SZ95 sebocytes in vitro.

Moringa seed oil induced a concentration-dependent increase in lipid synthesis in SZ95 sebocytes. The pattern of lipogenesis induced by Moringa seed oil was comparable to the lipogenesis pattern of SZ95 sebocytes obtained under maintenance with olive oil and the free fatty acids linoleic acid and oleic acid (Figure 1). Both fatty acids are known to cause lipogenesis in human sebocytes [16,17,18], while linoleic acid also induces a pro-inflammatory signaling [12]. At comparable concentrations sunflower oil induced a more prominent lipogenesis pattern than Moringa seed oil and olive oil (Figure 1). In quantitative comparison, the increase of intracellular lipid droplets induced by 10 µL Moringa seed oil resembled that induced by 100 µM linoleic acid.

The main component of Moringa seed oil is the fatty acid oleic acid, which is also found in comparable amounts in olive oil but in lower concentration in sunflower oil (Table 1). Oleic acid, a free fatty acid with numerous natural anti-inflammatory properties on sebocytes [19,20], caused strong lipogenesis in SZ95 sebocytes (Figure 1).

Enhanced lipogenesis is usually associated with cell death of lipid-producing cells [21]. Therefore, we investigated cell viability when treated with oils and fatty acids (Figure 2). The longitudinal cell viability experiments showed that the tested fatty acids, in concentrations similar to those detected in Moringa seed oil (Table 1), did not cause cell death, with only linoleic acid consistently leading to increased viability over a period of 6 days (Figure 2).

Interestingly, in addition to its more prominent lipogenesis pattern described above (Figure 1), sunflower oil induced a marked lipid synthesis in SZ95 sebocytes after a single administration over a period of 6 days (Figure 2).

### 2.3. Moringa Seed Oil Exhibits Anti-Inflammatory Properties

To investigate the biological significance of the marked difference in sebaceous lipogenesis caused by Moringa seed oil and the olive oil on one hand and sunflower oil on the other hand, we established a cytokine antibody array, which detects several cytokines simultaneously (Figure 3). Moringa seed oil, olive oil and oleic acid reduced the secretion of tumor necrosis factor (TNF)-α, tumor growth factor (TGF)-β and TNF-β in comparison to the untreated cells (and the plain culture medium) after 24 h of treatment (Figure 3). The detectable protein pattern with expression of IL-6, IL-8 and TGF-β—as a secondary finding—were a clear indication of the origin of the studied sebaceous gland cells as lipid-producing cells [15,19,22].

### 2.4. Moringa Oil Leads to an Anti-Inflammatory Fatty Acid Profile in Sebocytes

To further identify the possible anti- or proinflammatory effects of the studied vegetable oils, the uptake or synthesis of fatty acids in human SZ95 sebocytes after administration of the tested oils/fatty acids was investigated. Furthermore, the obtained fatty acid pattern was compared with the content of the studied oils. To detect the development of anti- or pro-inflammatory signaling pathways based on fatty acid metabolism, we analyzed the intracellular fatty acid concentrations 24 and 48 h after the addition of the studied oils/fatty acids in the culture medium (Table 2 and Table 3). Moringa seed oil- and olive oil-treated cells had comparable intracellular content of anti-inflammatory saturated fatty acids (SFA) to the control without oil treatment after 24 h, but after 48 h the SFA increased compared to the control SFA (Table 2). Under both conditions, cells showed higher amounts of oleic acid precursor stearic acid (C18:0) compared with the cells of the control. Cells treated with sunflower oil and linoleic acid showed lower levels of both SFA and stearic acid compared with the control. Finally, in fatty acids extracted from Moringa seed oil-treated cells the lowest n-6/n-3 ratio (pro-inflammatory/anti-inflammatory polyunsaturated fatty acids) was found, namely 1.24 after 24 h and 1.28 after 48 h, among all the oils studied, indicating the lowest pro-inflammatory activity, followed by olive oil—1.48 after 24 h and 1.59 after 48 h. In contrast, sunflower oil only exhibited a n-6/n-3 index of 2.42 after 24 h and 3.02 after 48 h (Table 2).

### 2.5. Effect of Moringa Seed Oil on SZ95 Sebocyte Proliferation

Application of all oils tested resulted in enhancements of cell proliferation compared to control (Figure 4). This proliferation enhancement decreased over time—in contrast to lipid synthesis—in a comparable manner by Moringa seed oil and oleic acid as well as olive oil and sunflower oil, while the delayed proliferation under linoleic acid resembled the control proliferation.

## 3. Discussion

The wide range of applications of *Moringa oleifera* in Africa and the similarities of Moringa seed oil with olive oil [23] has led the Dryer Stiftung in Berlin, Germany, to include the cultivation of *Moringa oleifera* to the foundation’s projects in Dano, in the north of Burkina Faso, as an ecological and highly productive agriculture project with the highest possible proportion of value added locally [24].

Our contribution to this project was the scientific evaluation of the effects of Moringa seed oil on the function of human sebocytes, the cells which are involved in numerous inflammatory skin diseases [25] and control skin homeostasis. Sebocytes have, therefore, been named “the brain of the skin” [26]. Moringa seed oil effects were compared to those of other popular vegetable oils, such as olive oil and sunflower oil, and to free fatty acids with important, characteristic patterns of action.

All vegetable oils and fatty acids induced sebocyte proliferation and sebaceous lipogenesis. In vivo, however, sebocytes die and burst after maturation and subsequent lipogenesis, releasing their content, the sebum, onto the skin surface after programmed holocrine secretion [21,27,28]. In vitro, sebocytes continue to proliferate independently of their lipid production [21]; only the pro-inflammatory fatty acid arachidonic acid (C20:4n-6) leads to increased cell death of human sebocytes [15,28]. However, the proliferation pattern of the cells treated with different vegetable oils and fatty acids in the current study was indicative of their biological properties. SZ95 sebocytes treated with Moringa seed oil proliferated similarly to those cells treated with the potential anti-inflammatory fatty acid oleic acid. Olive oil and sunflower oil exhibited similar proliferation patterns but induced different cytokine and lipid secretion patterns. The n-6/n-3 ratio under Moringa seed oil was 1.24 after 24 h and 1.28 after 48 h, 1.48 and 1.59 under olive oil and 2.42 and 3.02 under sunflower oil, respectively. Linoleic acid induced delayed sebocyte proliferation associated with enhanced differentiation and pro-inflammatory lipogenesis, as previously described [8,12,18,19]. The study of inflammatory responses after treatment with the oils showed that Moringa seed oil and olive oil, in contrast to sunflower oil, reduced the secretion of TNF-α, TNF-β and TGF-β. These cytokines are involved in the immune response of cells. TNF-α, a pro-inflammatory cytokine, is involved in extensive inflammation in follicular skin diseases, such as acne, and also supports proliferation and sebum release of sebocytes [29,30]. TNF-β (lymphotoxin-α) can activate the inflammatory environment in human chondrocytes and is involved in the development of rheumatoid arthritis [31]. TGF-β signaling regulates lipogenesis in human sebaceous gland cells [22,32]. These data are consistent with the findings of this study that sunflower oil-treated SZ95 sebocytes exhibited higher TNF-α secretion and higher lipogenesis than SZ95 sebocytes treated with Moringa seed oil or olive oil. Underlining this result, cells treated with Moringa seed oil had the lowest n-6/n-3 ratio after both 24 h and 48 h followed by olive oil. Cells treated with sunflower oil had the highest n-6/n-3 ratios and thus exhibited the strongest pro-inflammatory activity among the oils tested.

Vegetable oils and also Moringa seed oil only contain small amounts of free fatty acids, such as oleic acid [6,7], while fatty acids are mainly present as bound glycerol esters in these oils. Oleic acid is an important representative of the monounsaturated fatty acids and the predominant fatty acid in Moringa seed oil and olive oil (Table 1) [6]. It is an n-9-fatty acid due to the location of its double bond and can therefore be produced in sebaceous gland cells from stearic acid (C18:0). Oleic acid mainly occurs chemically bound in triglycerides in almost all natural (vegetable and animal) oils and fats. In contrast, linoleic acid, the main fatty acid in sunflower oil, can have both pro- and anti-inflammatory effects, with the latter one been predominant on human sebocytes [9,11,12].

A comparison of bioactive compounds and the characterization of fatty acid profile of hempseed oil, Moringa oil, echium oil, extra virgin olive oil and linseed oil already used or proposed to be used in functional food products showed adequate values for acidity and oxidation status [33]. Moringa oil was shown to be rich in oleic acid, while the sebaceous fatty acid squalene was only found in significant amounts in extra virgin oil. Therefore, Moringa oil and extra virgin olive oil were considered interesting sources of bioactive compounds having great potential for the food industry. In addition, these oils have great potential to be included in the formulation of functional ingredients for the delivery—among others—of omega-3 fatty acids and antioxidants.

A current comparison of Moringa seed oil from Egypt, Pakistan, South Africa and Thailand has shown a fatty acid content of olive oil type placing Moringa seed oil for consideration as a cooking oil amongst its other uses [5].

Interestingly, a staphylococcal lipase obtained from *Staphylococcus epidermidis S2* and isolated from sebaceous gland-rich areas on the human facial skin displays its highest activity in the hydrolysis of olive oil at 32 °C and pH 8 [34]. This lipase is stable up to 45 °C and within the pH range from 5 to 9 and has high activity against tributyrin, waste soybean oil and fish oil. Its microbial origin and biochemical properties 98–99% identical with that of olive oil may make this staphylococcal lipase isolated from facial sebaceous gland-rich skin at least partially responsible for the anti-inflammatory activity of olive oil on human sebocytes detected in this study and suitable for use as catalyst in the cosmetic, medicinal, food or detergent industries.

## 4. Materials and Methods

### 4.1. Cell Cultures and Reagents

After excluding Mycoplasma contamination, immortalized human SZ95 sebocytes [35] at 26–35 subcultures were maintained in Sebomed^®^ culture medium supplemented with 10% fetal bovine serum, 50 µg/mL gentamicin, 5 ng/mL epidermal growth factor (all Sigma-Aldrich, Taufkirchen, Germany) and 1 mM Ca^2+^ in a humidified atmosphere containing 5% CO_2_ at 37 °C. The medium was renewed every 2 days. Subconfluent cell cultures were treated with accutase (Millipore, Schwalbach, Germany) and were then propagated in culture medium as described above. 2.5 × 10^5^ SZ95 sebocytes were seeded in 6-well plates for visualization of lipid droplets using Nile Red (Kodak, Rochester, NY, USA) fluorescence as well as for cytokine detection via cytokine antibody array (Abcam, Cambridge, UK). 10^4^ SZ95 sebocytes were treated similarly for cell viability in 24-well plates (Nunc, Rochester, NY, USA). Moringa seed oil (kindly provided by the Dreyer Stiftung, Berlin, Germany), olive oil (Mazola, Elmshorn, Germany), sunflower oil (Kunella, Cottbus, Germany), linoleic acid and oleic acid (both Sigma-Aldrich) were administered to the cells in a working concentration of 10^−4^ M (in 0.1% dimethyl sulfoxide, final concentration). The SZ95 sebocytes were treated with Moringa seed oil, olive oil, sunflower oil and linoleic acid in flasks (T25; Nunc). Oils were directly added to the cell cultures without the application of vehicle(s). Untreated cells maintained for the same time periods were used as controls. After incubation for 24 h and 48 h, cell pellets proceeded to lipid extraction and subsequent determination of relative fatty acid concentration via gas chromatography.

### 4.2. Fatty Acid Content in the Vegetable Oils Tested

The fatty acid content analysis in the Moringa seed oil tested was performed by Agrolab LUFA (Kiel, Germany) and kindly provided by the Dreyer Stiftung. The fatty acid content analysis of olive oil and sunflower oil was obtained from the product descriptions (Table 1).

### 4.3. Cell Viability

Cell viability was determined by staining cells with calcein-AM (PromoCell, Heidelberg, Germany), which is converted to green fluorescent calcein by intracellular esterases in viable cells. Cells, grown and treated in 24-well plates, were dissociated from the culture plates with accutase and stained with 2.5 μg/mL calcein-AM at 37 °C for 1 h. Labeled cells were washed with phosphate-buffered saline without Ca^2+^ and Mg^2+^ and were measured by fluorescence activated cell sorting (FL2H). In the calcein assay, the total cell number does not contribute to the results; only the percentage of active cells in the remaining cell population is determined.

### 4.4. Cell Proliferation

For growth curves, cell confluence was continuously monitored by real-time cell analysis (RTCA, xCELLigence; Roche Diagnostics, Berlin, Germany). The technique is based on microelectrodes integrated in the bottom of each well of special 96-well E-plates. The electric impedance corresponds to the attached cell numbers. Some 5000 SZ95 sebocytes were seeded per microtiter well and were incubated with Moringa seed oil, olive oil, sunflower oil, linoleic acid and oleic acid. The impedance was determined up to 60 h after seeding in 15 min intervals.

### 4.5. Lipid Staining

Cells grown in 6-well plates (Nunc) were incubated with 1 μg/mL Nile Red dye (Sigma-Aldrich) for 20 min at room temperature, as previously described [36]. The cultures were then observed under a fluorescence microscope using a 450–500 nm bandpass exciter filter by light emission of >528 nm (Nile Red stain).

### 4.6. Cytokine Secretion

Full-length proteins were extracted from the culture supernatants after treatment with Moringa seed oil, olive oil, sunflower oil, linoleic acid and oleic acid using the Qproteome FFPE Tissue Kit (Qiagen, Hilden, Germany). Protein concentration was determined using the Biorad Protein Assay (Bio-Rad, Hercules, CA, USA). For protein quantification, proteins were blotted with a Human Cytokine Antibody Array (ab133996; abcam, Cambridge, UK). The levels of 23 inflammatory cytokines (GCSF, GM-CSF, GRO (αβγ), GRO-α, IFN-γ, IL-1α, IL-2, IL-3, IL-5, IL-6, IL-7, IL-8, IL-10, IL-13, IL-15, MCP-1, MCP-2, MCP-3, MIG, RANTES, TGF-β, TNF-α, TNF-β) were simultaneously measured. The blots were imaged via a chemiluminescent imager (Fusion FX7 imaging system; Peqlab, Erlangen, Germany). Densitometric analysis was performed by Image J (version 1.54d; 30 March 2023) [37].

### 4.7. Lipid Extraction

Lipid extraction was performed adapted from Bligh and Dyer extraction [38] and according to established protocols [39,40]. Cells stored at −80 °C were thawed at room temperature, and 10 µL butylated hydroxytoluene (Roth, Karlsruhe, Germany) and ethylene diamine tetraacetic acid (Sigma, Steinheim, Germany) (0.02 mg/mL each) in methanol (Merck, Darmstadt, Germany)/water 1:1 were added as antioxidant. This was followed by the addition of 500 µL ice cold methanol, 500 µL dichloromethane (Roth) and 250 µL water. Zirconium beads (1.5 mm; Biozym, Hessisch Oldendorf, Germany) were added to the mixture, and then the cells were homogenized for 1 min (BeadBug 3 Homogenizer; Biozym). After centrifugation (3500 rpm, 10 min, 4 °C), the lower phase was collected, and a second extraction was performed after adding 250 µL dichloromethane. The dichloromethane phases were combined and evaporated for 45 min at 37 °C in a vacuum concentrator.

### 4.8. Fatty Acid Determination

Fatty acid methylation and preparation was carried out according to an established protocol [16]. To prepare the derivatization (methylation) of fatty acids, the dried lipid extracts were mixed with 500 µL boron trifluoride in 14% methanol (Sigma-Aldrich) and 500 µL n-hexane (Roth) in glass vials and tightly closed. After vortexing, the samples were incubated for 60 min in a preheated block at 100 °C. After cooling down to room temperature, the mixture was added to 750 µL water, vortexed and shaken out for 4 min. Then, all samples were centrifuged for 5 min. (4 °C, 3500 rpm). From each sample, 100 µL of the upper n-hexane layer was transferred into a micro-insert in a glass vial, tightly closed and measured by gas chromatography.

### 4.9. Gas Chromatography

Gas chromatography was performed on a 7890B GC System (Agilent, Santa Clara, CA, USA) with a HP88 column (112/8867, 60 × 0.25 mm × 0.2 µm) for 30 min with the following temperature gradient: 50 °C to 150 °C with 20 °C/min, 150 °C to 240 °C with 6°C/min and 240°C for 10 min. Nitrogen was used as carrier gas (constant flow 1 mL/min). A total of 1 µL of the samples was injected into the injector (splitless injection, 280 °C). Flame ionization detection was performed at 250°C with the following gas flows: hydrogen 20 mL/min, air 400 mL/min and make up 25 mL/min. Methylated fatty acids in the samples were identified by comparing the retention times with those of known methylated fatty acids of the Supelco^®^ 37 FAME Mix and single FAME standards purchased from Cayman Chemicals (Ann Arbor, MI, USA). Analysis and integration of the peaks was carried out with the OpenLAB CDS ChemStation (Agilent).

### 4.10. Statistical Analysis

All results are presented as mean ± standard deviation. Statistical significance between two stimulation conditions was determined using the Wilcoxon matched-pairs signed-rank test. Statistical analysis with more than two stimulation conditions was performed with the Kruskal–Wallis test and Dunn’s multiple comparison test to correct for multiple testing. Asterisks represent statistical significance and are defined as * *p* < 0.05, ** *p* < 0.01, *** *p* < 0.005 and **** *p* < 0.001. If no asterisk is given or “ns” is mentioned, no statistical differences could be detected.

## 5. Conclusions

Olive oil is currently administered in several skin care indications, including skin hygiene, the protection of neonatal skin, skin burns, the treatment of grade I pressure ulcers and the management of acute radiation dermatitis [41,42,43,44,45,46]. Moringa seed oil apparently combines several desired properties which are comparable with olive oil and could, therefore, be a candidate for further investigations aimed with the intention of clinical application, both as a nutrient and ingredient of skin care. It contains high concentrations of the potential anti-inflammatory fatty acid oleic acid. It affects and increases the proliferation and lipogenesis of human sebocytes in a matter similar to oleic acid. Furthermore, it induces anti-inflammatory cytokine and fatty acid profiles in human sebocytes.

## Figures and Tables

**Figure 1 ijms-24-10332-f001:**
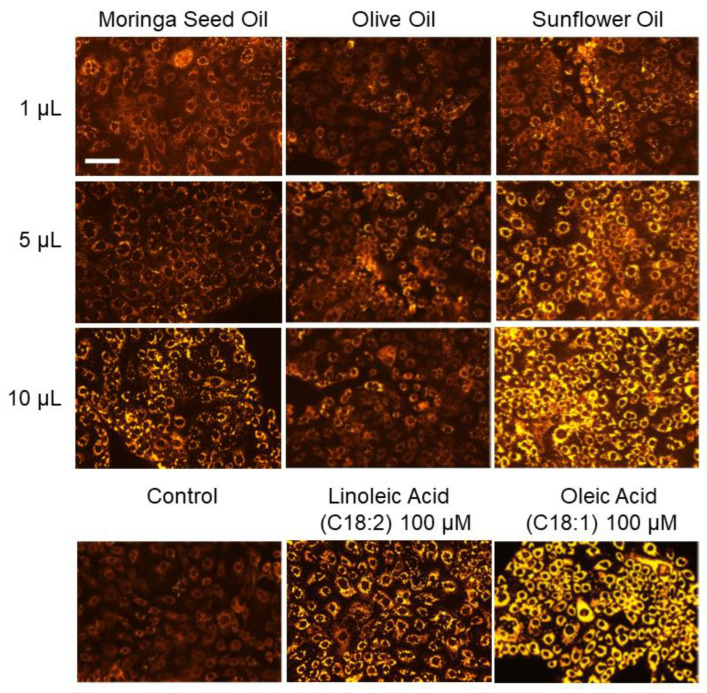
Intracellular accumulation of lipid droplets in the cytoplasm of human SZ95 sebocytes in single-layer cell culture treated with Moringa seed oil (oleic acid concentration 2.3 × 10^−5^ M), olive oil, sunflower oil and the free fatty acids linoleic acid and oleic acid and shown by Nile Red fluorescence. The figures are representative of five independent experiments with similar results. Scale = 10 mm.

**Figure 2 ijms-24-10332-f002:**
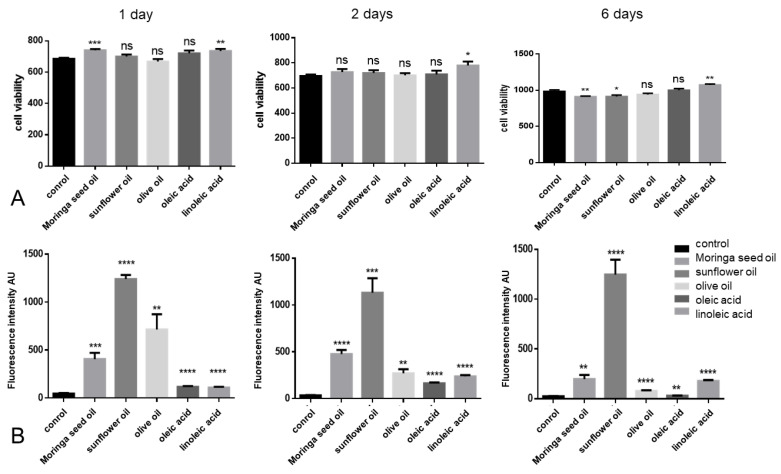
(**A**) Flow cytometry measurement of the esterase activity (detected by calcein-AM) representing cell viability (y axis: cell viability arbitrary units (AU)) of SZ95 sebocytes treated with different oils/lipids for up to 6 days. The figures are representative of three independent experiments with similar results. * *p* < 0.05, ** *p* < 0.01, *** *p* < 0.005, ns = non-significant. (**B**) Longitudinal assessment of intracellular neutral lipid content (y axis: Fluorescence intensity arbitrary units (AU)) in SZ95 sebocytes treated with different oils/lipids for up to 6 days. The figures are representative of at least three independent experiments with similar results. ** *p* < 0.01, *** *p* < 0.005, **** *p* < 0.001.

**Figure 3 ijms-24-10332-f003:**
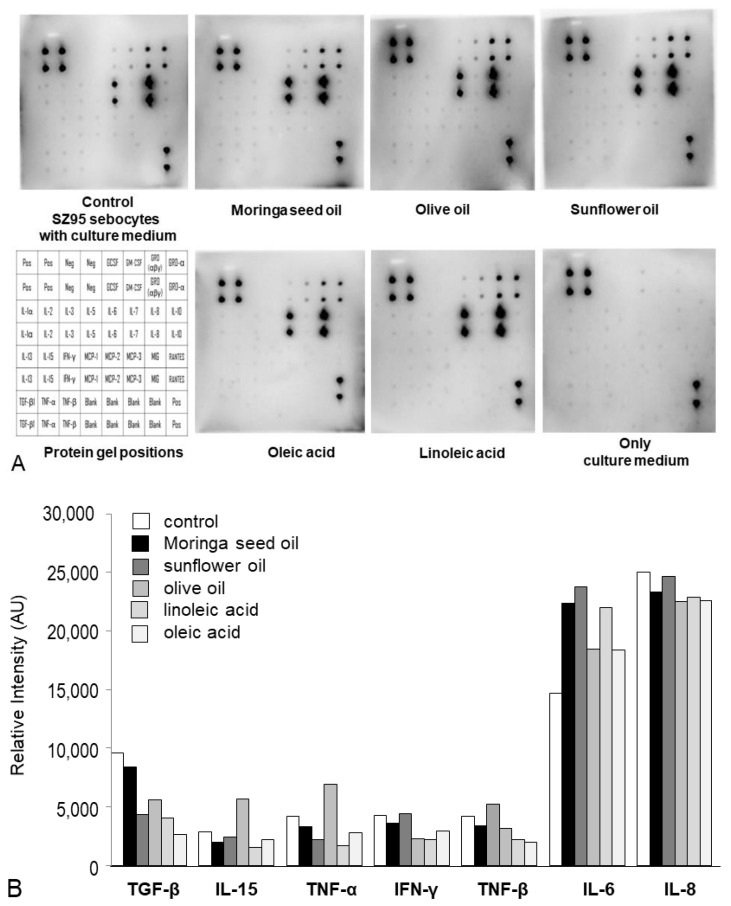
Cytokine secretion pattern of SZ95 sebocytes assessed in the culture supernatants after 24 h-treatment with different oils/lipids. (**A**) Full-length proteins were extracted from the culture supernatants after treatment with Moringa seed oil, olive oil, sunflower oil, linoleic acid and oleic acid and quantified as described in Section 4. Supernatant proteins were then blotted with a Human Cytokine Antibody Array of 23 inflammatory cytokines with a double arrangement on the blot as shown at the lower left part of the figure (Pos = positive, Neg = negative, Blank = no protein, GCSF = granulocyte colony-stimulating factor, GM-CSF = granulocyte macrophage-colony-stimulating factor, GRO = growth-regulated protein, IL = interleukin, IFN = interferon, MCP = monocyte chemoattractant protein, MIG = monokine-induced by IFN-γ, RANTES = regulated and normal T cell expressed and secreted, TGF = tumor growth factor, TNF = tumor necrosis fector). The blots were imaged via a chemiluminescent imager. (**B**) Densitometric analysis was performed by Image J (version 1.54d; 30 March 2023). The culture medium without cells and the culture medium of SZ95 sebocytes served as controls. The relative arbitrary units of the signals from immunoblots conducted in a separate, representative experiment are shown.

**Figure 4 ijms-24-10332-f004:**
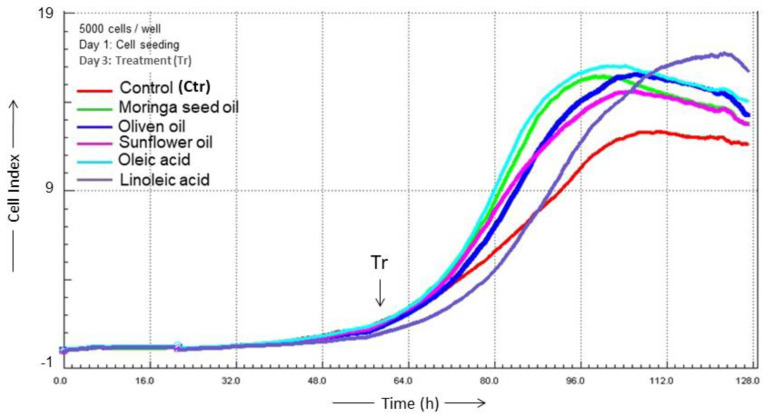
Real time cell analysis of proliferation of SZ95 sebocytes (5000 cells/well) under treatment with different oils/lipids and impedance detection in gold-coated culture plates. SZ95 sebocyte growth curves were determined in real time (×CELLigence). The y axis shows the impedance exhibited by the number of cells attached at the bottom of the well. Cells were treated for 80 h after seeding with Moringa oil (2.3 × 10^−7^ M), olive oil (10 µL), sunflower oil (10 µL), linoleic acid (10^−4^ M) and oleic acid (10^−4^ M). Effects were compared to non-treated controls (Ctr). Time of treatment is indicated by a vertical arrow (Tr): day 3 after seeding the cells). The curves are representative of two independent experiments with similar results.

**Table 1 ijms-24-10332-t001:** Distribution (in %) of the major fatty acids in the vegetable oils tested. Since the numbers are average values and do not represent the entire content of fatty acids, the addition of the data for each oil does not obligatorily add up to 100%.

Fatty Acid	Moringa Seed Oil	Olive Oil	Sunflower Oil
Saturated fatty acids	23	15	9
Monounsaturated fatty acids	75	70	27
Polyunsaturated fatty acids	1	6	65
Oleic acid (C18:1n-9)	66	72	14–37
Palmitic acid (C16:0)	6	11	5–8
Linoleic acid (C18:2n-6)	1	8	48–74
Stearic acid (C18:0)	6	2	3–6
Palmitoleic acid (C16:1n-7)	2	0–4	

**Table 2 ijms-24-10332-t002:** Fatty acid composition of human SZ95 sebocytes after incubation with the investigated oils or fatty acids (24 h and 48 h; representation in relative fatty acid concentrations [%]) (Σ = sum; SFA = saturated fatty acids, MUFA = monounsaturated fatty acids, PUFA = polyunsaturated fatty acids).

Fatty Acids	Control		Moringa Seed Oil		Olive Oil		Sunflower Oil		Linoleic Acid	
	24 h	48 h	24 h	48 h	24 h	48 h	24 h	48 h	24 h	48 h
SFA	31.9	20.0	26.5	29.1	30.4	25.7	13.6	15.7	16.8	14.2
C12:0	-	-	-	-	-	-	0.17	0.1	-	-
C13:0	0.8	0.5	0.2	0.5	0.2	0.3	0.2	0.1	0.1	0.1
C14:0	0.7	0.4	0.6	0.5	0.6	0.5	0.3	0.4	0.5	0.5
C16:0	20.3	12.7	14.2	15.1	16.4	13.6	7.3	8.2	8.7	6.9
C17:0	2.5	1.9	1.4	1.8	1.6	1.5	1.0	0.8	0.7	0.7
C18:0	7.6	4.5	8.5	9.0	9.7	8.6	3.9	5.6	6.3	5.6
C24:0	0	0	1.5	2.2	1.8	1.3	0.7	0.4	0.5	0.5
ΣMUFA	48.1	55.3	56.6	53.0	50.0	54.3	65.8	65.7	20.2	14.5
C16:1n-7	2.1	2.7	2.2	2.5	2.1	1.9	1.2	0.8	0.9	0.7
C18:1n-9t	8.4	1.2	0.5	2.3	-	-	0.5	0.3	0.3	0.3
C18:1n-9c	32.5	44.5	45.8	39.6	40.8	45.5	59.4	60.7	15.8	10.9
C18:1n-7	5.1	7.0	6.2	6.0	5.4	5.2	3.2	2.3	2.8	2.1
C20:1n-9	0	0	0.8	1.2	0.8	0.7	0.7	1.0	0	0
C24:1n-9	0	0	1.1	1.5	1.0	1.1	0.8	0.6	0.4	0.5
ΣPUFA	20.1	24.7	16.9	17.9	19.6	20.0	20.6	18.6	63.1	71.3
Σn-3-PUFA	9.1	12.1	7.6	7.9	8.1	7.7	5.8	4.4	4.5	4.0
C20:5n-3	1.9	1.7	1.0	1.2	1.0	1.1	0.6	0.5	0.6	0.5
C22:5n-3	2.7	3.8	2.7	2.7	2.7	2.6	2.3	1.7	1.6	1.5
C22:6n-3	4.5	6.6	3.9	4.0	4.3	4.0	2.9	2.3	2.3	2.0
Σn-6-PUFA	10.9	12.6	9.4	10.0	11.5	12.3	14.1	13.4	58.6	67.3
C18:2n-6c	2.9	2.1	1.7	2.0	2.4	3.5	5.6	6.0	43.7	52.9
C18:3n-6	0	0	0	0	0	0	0.5	0.4	1.6	1.3
C20:2n-6	0	0	0	0	0	0	0	0	1.3	2.1
C20:3n-6	0	0	0	0	0.9	0.8	0.7	0.6	2.4	2.2
C20:4n-6	8.0	10.5	7.7	8.1	8.3	8.0	6.5	5.7	8.5	7.6
C22:4n-6	0	0	0	0	0	0	0.9	0.7	1.2	1.4
Σn-9-PUFA										
C20:3n-9	0	0	0	0	0	0	0.7	0.8	0	0
n-6/n-3	1.20	1.04	1.24	1.28	1.43	1.59	2.42	3.02	13.11	16.81

**Table 3 ijms-24-10332-t003:** Codes of the investigated fatty acids.

Saturated Fatty Acids		Monounsaturated Fatty Acids		Polyunsaturated Fatty Acids	
C12:0	Lauric acid				
C13:0	Tridecanoic acid				
C14:0	Myristic acid				
C16:0	Palmitic acid	C16:1n-7	Palmitoleic acid		
C17:0	Margaric acid				
C18:0	Stearic acid	C18:1n-9t	Elaidic acid	C18:2n-6c	Linoleic acid
		C18:1n-9c	Oleic acid	C18:3n-6	γ-Linolenic acid
		C18:1n-7	cis-Vaccenic acid		
		C20:1n-9	11-Eicosenoic acid	C20:2n-6	11,14-Eicosadienoic acid
				C20:3n-6	homo-γ-Linolenic acid
				C20:3n-9	5,8,11-Eicosatrienoic acid (Mead acid)
				C20:4n-6	Arachidonic acid
				C20:5n-3	Eicosapentaenoic acid
				C22:6n-3	Docosahexaenoic acid
C24:0	Lignoceric acid	C24:1n-9	Nervonic acid		

## Data Availability

The data presented in this study are available on request from the corresponding author.

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
