# Peer review of "Effects of Moringa oleifera Seed Oil on Cultured Human Sebocytes In Vitro and Comparison with Other Oil Types"

_ijms, 2023, doi:10.3390/ijms241210332_

Round 1

Reviewer 1 Report

The topic is interesting not only from the scientific viewpoint but exhibits also non-negligible social background. The comparison presented for various oil types provides the interesting conclusions. However, some parts should be improved for better understanding of the results:

- ll. 93-95: very short and insufficient formulation of the aim;

- Table 1: How to understand the numbers?

- Fig. 1: The scale is missing.

- Fig. 2: For better visual comparison all three figures should be  condensed to the only one depicting 6 triples, each triple with three adjacent columns (1, 2, 6 days).

- Fig. 3: as in Fig. 2 – 6 triples;

- Table 2: accuracy of the individual numbers (LA)?

- Conclusions: very general sentences, no comparison with other products, no advantages.

Just for improvement:

- l. 49: increase;

- l. 162: pro-inflammatory;

- l. 223: -4;

- l. 275: 1´104;

- l. 331: centrifugation

Author Response

We thank the reviewer for his extremely careful evaluation, also considering the background of our work, which represented a “donation” to the Dryer Stiftung supporting its continuous dedication in Africa.

Lines 93-95: We have reformulated the aim of our work, as follows: “In this work, we assessed the effects of Moringa seed oil on human SZ95 sebocytes and compared them with the effects of other vegetable oils, namely olive oil and sunflower oil, as well as with the pure fatty acids LA and oleic acid. Our aim was to assess any regulatory properties of Moringa seed oil on the physiology of human sebocytes in order to preclinically evaluate a possible application of this oil as an ingredient for cosmetics and a topical drug vehicle in addition to its use as a nutrient in Africa.”

Table 1: As mentioned in the Table title, the numbers represent the average percentage of the major fatty acids in the oils tested. Since the numbers are average values and not the entire contant of fatty acids is presented, the addition of the presented data for each oil does not obligatorily result to 100%. This is now mentioned at the Tavle title.

Fig. 1: A scale is now added to the figure and mentioned in the figure legend.

Fig. 2 and 3 have been condensed to the new figure 2. Consequently figures 4-5 bcame figures 3-4.

Table 2: We cordially thank the reviwer for the detection of the wrong transfer of the LA data in the table. The mistakes at the LA results were corrected and all results double chcked for any additional transfer mistakes.

The “Conclusions” section has been rewritten to address the queries of the reviewer and reads now: “Olive oil is currently administered in several skin care indications, including skin hygiene, the protection of neonatal skin, skin burns, the treatment of grade I pressure ul-cers and the management of acute radiation dermatitis [41–46]. Moringa seed oil appar-ently combines several desired properties which are comparable with olive oil and could, therefore, be a candidate for further investigations aimed with the intention of clinical application, both as a nutrient and ingredient of skin care: It contains high con-centrations of the potential anti-inflammatory fatty acid oleic acid. It affects and increas-es the proliferation and lipogenesis of human sebocytes in a matter similar to oleic acid. Furthermore, it induces anti-inflammatory cytokine and fatty acid profiles in human sebocytes.”

Line 49: increase; line 223: -4; line 275: 1´104; line 331: centrifugation: The wrong terms were corrected.

Line 162: The more precise term here is “lowest pro-inflammatory activity”

Reviewer 2 Report

The presented work gives interesting results of the potential application of Moring seed oil.

1. It is worth repositioning the "Material and methods" section. Currently, this section is located at the end of the article before the conclusion, and they are placed after the results obtained. 2. The pictures presented between lines 198 and 199 are not labeled and it is not clear what they are about. 3. It is worth expanding the information on the potential use of such a preparation in humans. Currently, there is little of it, and its value is emphasized in the conclusions.  4. References should be expanded.  

Author Response

We thank the reviewer for his motivating positive comments on our work.

  1. The section “Materials and methods” follows the “Discussion” due to the format of the journal, which rquires this sequence.
  2. Figure 4 is now labeled as A and B and the legend includes more details explaining the two parts of the figure: “Figure 4. Cytokine secretion pattern of SZ95 sebocytes assessed in the culture su-pernatants after 24 h-treatment with different oils/lipids. A. Full-length proteins were extracted from the culture supernatants after treatment with Moringa seed oil, olive oil, sunflower oil, linoleic acid and oleic acid and quantified as described in the Material and methods. Supernatant proteins were then blotted with a Human Cytokine Antibody Ar-ray of 23 inflammatory cytokines with a double arrangement on the blot as shown at the lower left part of the figure. The blots were imaged via a chemiluminescent imager. B. Densitometric analysis was performed by Image J. The culture medium without cells and the culture medium of SZ95 sebocytes served as controls. The relative arbitrary units of the signals from immunoblots conducted in a separate, representative experiment are shown.”
  3. We have tried to avoid speculations regarding the use of Moringa oil in humans only based on in vitro data. However, we followed the query of the reviewer and expanded the information in the “Discussion” section and in the “Conclusions”. These paragraphs are added in the Discussion

“A comparison of bioactive compounds and the characterization of fatty acid profile of hempseed oil, moringa oil, echium oil, extra virgin olive oil and linseed oil already used or proposed to be used in functional food products showed adequate values for acidity and oxidation status [33]. Moringa oil was shown to be rich in oleic acid, while the sebaceous fatty acid squalene was only found in significant amounts in extra virgin oil. Therefore, moringa oil and extra virgin oil were considered interesting sources of bi-oactive compounds having great potential for the food industry. In addition, these oils have great potential to be included in the formulation of functional ingredients for the delivery – among others - of omega-3 fatty acids and antioxidants.

A current comparison of Moringa seed oil from Egypt, Pakistan, South Africa and Thailand has shown a fatty acid content of olive oil type placing moringa seed oil for consideration as a cooking oil amongst its other uses [34].

Interestingly, a staphylococcal lipase obtained from Staphylococcus epidermidis S2 and isolated from sebaceous areas on the human facial skin displays its highest activity in the hydrolysis of olive oil at 32°C and pH 8 [35]. This lipase is stable up to 45° C and within the pH range from 5 to 9 and has high activity against tributyrin, waste soybean oil and fish oil. Its microbial origin and biochemical properties 98-99% identical with that of olive oil may make this staphylococcal lipase isolated from facial sebaceous skin at leasr partially responsible for the anti-inflammatory activity of olive oil on human sebocytes detected in this study and suitable for use as catalyst in the cosmetic, medici-nal, food or detergent industries.”

and the Conclusion has been enriched
“Olive oil is currently administered in several skin care indications, including skin hygiene, the protection of neonatal skin, skin burns, the treatment of grade I pressure ul-cers and the management of acute radiation dermatitis [41–46]. Moringa seed oil appar-ently combines several desired properties which are comparable with olive oil and could, therefore, be a candidate for further investigations aimed with the intention of clinical application, both as a nutrient and ingredient of skin care: It contains high con-centrations of the potential anti-inflammatory fatty acid oleic acid. It affects and increas-es the proliferation and lipogenesis of human sebocytes in a matter similar to oleic acid. Furthermore, it induces anti-inflammatory cytokine and fatty acid profiles in human sebocytes.”.

  1. References were adequately expanded.

Round 2

Reviewer 1 Report

The authors addressed all the comments raised in the preceding review.